# Impact of the Technical Snow Production Process on Bacterial Community Composition, Antibacterial Resistance Genes, and Antibiotic Input—A Dual Effect of the Inevitable

**DOI:** 10.3390/ijms26062771

**Published:** 2025-03-19

**Authors:** Klaudia Stankiewicz, Klaudia Bulanda, Justyna Prajsnar, Anna Lenart-Boroń

**Affiliations:** 1Department of Microbiology and Biomonitoring, Faculty of Agriculture and Economics, University of Agriculture in Kraków, Adam Mickiewicz Ave. 24/28, 30-059 Kraków, Poland; klaudia.kulik@student.urk.edu.pl; 2Department of Forest Ecosystems Protection, Faculty of Forestry, University of Agriculture in Kraków, 29 Listopada Ave. 46, 31-425 Kraków, Poland; klaudia.bulanda@student.urk.edu.pl; 3Jerzy Haber Institute of Catalysis and Surface Chemistry, Polish Academy of Sciences, Niezapominajek Str. 8, 30-239 Kraków, Poland; justyna.prajsnar@ikifp.edu.pl

**Keywords:** antibiotics, antibiotic resistance genes, bacterial community composition, metataxonomy, next-generation sequencing, ski stations, storage reservoirs, technical snow

## Abstract

Although climate warming-induced snow cover reduction, as well as the development of ski tourism in hot and dry countries, is shifting industries toward the use of technical snowmaking, its use raises hydrological, health-related, and environmental concerns. This study was aimed at enhancing our current understanding of the impact of technical snowmaking on the environment and human health. Culturable bacteriological indicators of water quality (*Escherichia coli*, fecal enterococci, *Salmonella*, and *Staphylococcus*), the presence and concentration of antimicrobials, genes determining bacterial antibiotic resistance (ARGs), and next-generation sequencing-based bacterial community composition and diversity were examined from river water, technological reservoirs, and technical snow from five ski resorts. The number of culturable bacteria and prevalence of most ARGs decreased during snowmaking. The concentration of antimicrobial agents changed irregularly, e.g., ofloxacin and erythromycin dropped in the snowmaking process, while cefoxitin was quantified only in technical snow. The bacterial community composition and diversity were altered through the technical snowmaking process, resulting in the survivability of freezing temperatures or the presence of antimicrobial agents. Water storage in reservoirs prior to snowmaking allows us to reduce bacterial and ARG contaminants. Frequent and thorough cleaning of snowmaking devices may aid in reducing the negative impact snowmaking can have on the environment by reducing contaminant input and limiting the disturbance of the ecological balance.

## 1. Introduction

Climate warming-induced snow cover reduction results in technical (artificial) snow production becoming inevitable in moderate climate regions. The all-weather snowmaking systems [1] that have been developed and strongly advertised in recent years have allowed the production of technical snow in warm and dry climate regions. In 2023, Saudi Arabia, which is a typical hot desert country, announced the construction of a ski village, Trojena, where technical snow will be produced in large amounts. The technical snow is produced as a surrogate for natural snow in the region, where natural snow is either missing or unpredictable, and many ski runs rely entirely on technical snow for part of or the entire season.

Mountain areas worldwide are subjected to significant anthropogenic pressure, as they are visited by numbers of tourists that, in many cases, exceed the numbers of permanent residents by several times [2,3]. Such intensive tourism generates wastewater that often exceeds the capacities of local treatment plants, resulting in the contamination of surface water systems [4]. Importantly, most wastewater treatment plants (WWTPs) in many countries are not efficient in the removal of some pollutants, such as antimicrobial agents or antibiotic resistance genes, which makes them not only significant environmental sources of these pollutants in aquatic environments [5] but also hotspots of antimicrobial resistance. This results in surface water resources in mountain areas being commonly and severely contaminated by the above-mentioned pollutants and micropollutants, hence the increasing concerns that are associated with the effects of technical snowmaking on human health and the environment. The contaminants of water resources, such as potentially pathogenic bacteria, antibiotic-resistant bacteria (ARB), and antibiotic resistance genes (ARGs), as well as antimicrobial agents, once present therein, can be transferred into technical snow and possibly re-enter the environment via meltwater runoff [6,7]. In addition, human activities have established a water circulation process in which treatment and discharge of treated wastewater, water uptake, treatment, and distribution are key stages [8]. Thus, once hardly eliminated contaminants enter this cycle, they may never be removed entirely.

Apart from the meteorological and hydrological aspects, as well as water quality and health-related concerns of technical snow production, another aspect to consider is the ecological impact it can have on soil at various levels. These issues include soil insulation, altered temperature conditions, mineralization increase due to snowmelt inflow [6,7], and finally, alteration of soil microbial communities, as well as horizontal transfer of antibiotic resistance genes between pathogenic and environmental bacteria. To date, the scientific literature lacks studies concerning the effects of technical snow production related to its possible input of bacteria and micropollutants, such as antimicrobial agents, antibiotic-resistant bacteria, and genetic determinants of antimicrobial resistance. The few studies that consider potential hazards related to bacteria associated with technical snow concern bacterial additives used as ice nucleation activators, *Pseudomonas syringae* [9]. Bearing in mind the fact that technical snowmaking will become an increasingly important aspect of tourism, not only in countries with a moderate climate but also in hot and dry areas, the issues associated with technical snow production will become important on a global scale. For this reason, the current lack of studies on both the health-related and environmental impacts of technical snowmaking seems incomprehensible and worth changing.

Undoubtedly, the process of technical snow production does not remain indifferent to the adjacent environment, and the aspects in which it can affect the ecosystem are multiple, diverse, and require interdisciplinary analysis. Thus, this study was undertaken in order to expand the currently limited knowledge of environmental and health-related impacts caused by technical snow production. This research was based on resource water (river water and reservoir-stored water) and technical snow (analyzed as snowmelt water) collected from five ski resorts located in the Polish mountains; these differed in height above sea level, population size, and distance from pollution sources. A number of parameters were examined, i.e., culturable bacteriological indicators of water quality, presence and concentration of antimicrobial agents, genes determining bacterial antibiotic resistance, as well as next-generation sequencing-based bacterial community composition and diversity.

By analyzing the above-listed parameters, each one separately as well as all together, we aimed to assess whether and how the process of technical snow production using water of varying quality and containing different concentrations of antibacterial agents changes the above-listed parameters. Information gained in this study will allow us to elucidate the possible effects technical snowmaking can have on the aquatic and soil environment in the close proximity of ski stations. By analyzing interrelations between the examined parameters, as well as their differences between the ski stations and throughout the snowmaking process, we aimed to select the factors that contribute to either elimination or further spread of contaminants in the environment. Finally, by assessing the changes in bacterial community composition and diversity through the technical snowmaking process, we aimed at determining the impact it can have on the ecological balance of bacterial communities in aquatic and soil environments in the direct vicinity of ski stations.

## 2. Results and Discussion

### 2.1. Culture-Based Assessment of Bacteriological Contamination

The culture-based analysis of bacterial contamination indicators (i.e., *Escherichia coli*, fecal enterococci, coagulase-positive *Staphylococcus*, and *Salmonella*) was used as the first stage assessment and arrangement of the examined stations in terms of the overall quality. It is evident that numbers of culturable bacteria vary between the sites situated in different regions and between the samples of water/snow, as well as water/reservoir/snow (Table 1; detailed parameters given in Appendix A; the names of regions and sampling sites are given in acronyms, which are explained in the Section 3 Table 4). The most evident differences can be seen in the case of *E. coli*, the mean numbers of which range from its absence in the presumably most pristine region (BDF) to more than 224 × 10^3^ CFU/100 mL at BDZW (river water), and this difference is statistically significant (F = 5.250, *p* = 0.001; Appendix A_statisical analysis). Also, the numbers of fecal enterococci were the lowest at the BDF site and the highest at BDZ (Table 1). These evident differences seem to be affected by both the number of inhabitants (the smallest at the BDF site and the highest at BDZ, Table 4) and the distance of water intake from the potential point sources of pollution, such as wastewater treatment plants (WWTPs) and hospitals, the effluents of which are not pre-treated before being discharged into the sewerage system. In our earlier study [10], where the technical snow contamination was preliminarily assessed, the river water intake sites situated downstream of the WWTPs were most severely contaminated. Similar observations have been reported in many countries worldwide, e.g., by Hyer et al. in their study conducted in Shenandoah National Park in Virginia, US [11], or Lu et al. [12], who found significant alterations in bacterial community composition as a result of wastewater effluent being discharged into receiving river water. Also, hospital effluents or proximity of hospitals have been demonstrated as important point sources of aquatic environment pollution with bacterial contaminants [13] and antibiotic-resistant bacteria [14,15]. On the other hand, bacteriological contaminants in less impacted regions, such as BDF or B3, situated nearby the Tatra National Park in Poland and Slovakia (Europe), were still present, but their numbers were very low (Table 1). In such places, similarly to the study by Maes et al. [16], conducted on an oligotrophic river in northern Sweden, there are often no obvious relations between point sources of pollution and the diffuse sources of pollution that are possible, e.g., human fecal pollution from trails where no dry toilets are available or fecal pollution by wild animals.

At most sites, the numbers of bacterial contaminants sharply drop during the technical snow production process. The similar drops in these values were observed in the case of fecal enterococci, *Salmonella*, and *S. aureus* (Table 1). Staley et al. [17], in their lab-designed microcosm experiment, in which snow kept at −10 and +4 °C was inoculated with *E. coli* strains or raw wastewater, demonstrated that while bacterial contaminants significantly decreased in low temperatures, those that survived could persist for weeks or months. Interestingly, the mean numbers of *Staphylococci* and *Enterococci* did not drop as evidently as the numbers of *E. coli* (Table 1). Among possible explanations is the ability of these bacteria to form biofilms on various solid surfaces that allow them to survive harsh conditions [18] or even recover from inactivation [19]. When biofilm sections periodically break off, they enter the bulk water and cause a sudden increase in bacterial counts [20].

### 2.2. Concentration of Antimicrobial Agents

Out of the 21 antimicrobial agents examined, 14 were detected above the levels of quantification (LOQ) (Table 2). A diverse range of the mean concentrations of antimicrobial agents was observed across the studied sites. With respect to the antimicrobial agents, the three most frequently detected antimicrobials comprised erythromycin (detected in 71% of all examined samples), tetracycline, and trimethoprim (both detected in 35.5% of samples). However, these antimicrobials were not among the ones whose highest concentrations were observed during the study, like ofloxacin, the total concentration of which exceeded 2000 ng/L.

Regarding the differences between studied sites in terms of the antibiotic loads and prevalence, the more anthropogenically impacted sites, i.e., R and BDZ, were characterized by both the highest total concentrations of antimicrobials and the highest detection rate (Table 2). The concentration of sulfamethoxazole at the BDZ and R sites differed significantly compared to other sites (F = 2.649, *p* = 0.03). The frequency of antibiotic detection within the samples is associated with the intake sites’ distance from the point sources of pollution, such as WWTPs and hospitals, which is similar to what other authors have demonstrated, who examined the contamination of aquatic environments with antimicrobial agents [13,21,22]. The detection rates of individual antibiotics might be influenced by the seasonal patterns of respiratory infections in temperate climates and the prescription rates of e.g., macrolide antibiotics (such as erythromycin) or combinations of trimethoprim/sulfamethoxazole [23,24].

The quality of water in three out of the five examined sites was impacted by wastewater from various point sources: raw domestic effluents, effluents from nearby WWTPs, and hospitals [21,25,26]. The concentrations of antibiotics in wastewater effluents were influenced by a number of factors, including pattern and consumption rate, excretion rates, and efficacy of elimination during the treatment process [27]. An important factor affecting the detection rates of antimicrobials was their persistence in aqueous environments, and such are the ones detected most frequently in our study (i.e., macrolide erythromycin, and trimethoprim) as well as fluoroquinolone [28] ofloxacin, the total concentration of which was the highest among all antimicrobials detected. Interestingly, in a few cases, the concentrations of antimicrobial agents in technical snowmelt water exceed the ones observed in river water or in reservoir-stored water. On the other hand, the concentrations of all antimicrobials in the R site were significantly lower (or none) in snowmelt water than in river water. There may be a few possible explanations for this irregularity. First of all, largely fluctuating concentrations of antimicrobials in flowing river water due to their fluctuations in wastewater effluents [24], which resulted in the antimicrobials not being captured during river water sampling, while the technical snow production resulted in significant volumes of water being trapped and stagnating on the slope surfaces. Secondly, the technical snow production process involves sublimation and evaporation while water droplets are traveling through the air, thus resulting in certain amounts of water being lost (hence its high density and degree of compaction) [6,29], but not the compounds suspended therein. Finally, the degradation rates of antimicrobial agents vary depending on the environmental conditions. And so, following kinetic theory, lower temperatures may slow down the process of antibiotics’ decomposition as the number of molecular collisions (affecting reactivity) increases as a function of temperature [30]. Also, Loftin et al. [31] observed longer half-lives (including tetracyclines) of antibiotics in lower temperatures.

Even though the concentrations of antibiotics detected in this study are low, it needs to be stressed here that the observed concentrations are subinhibitory and thus are associated with a number of adverse effects, including the following: chronic toxicity, alterations in bacterial community structure and the most dramatic, recent issue—the development of antibiotic resistance among bacteria following the environmental occurrence of antibiotic resistance genes [22,32]. Due to the multiple threats posed by the occurrence of antimicrobials in the environment, many countries have introduced regulations requiring the monitoring of certain substances. The EU Commission has established a watch list of substances for Union-wide monitoring under the Water Framework Directive. The selection of antibiotics for the watch lists is in line with the European One Health Action Plan against antimicrobial resistance [33]. The selection of antibiotics for the fourth watch list includes clindamycin and ofloxacin [33], both detected in 29% of samples in this study, and both of these substances are suggested to pose a risk to the aquatic environment. Importantly, the total concentration of ofloxacin in this study was the highest among all antimicrobials (i.e., 2030.53 ng/L, Table 2).

Most of the antimicrobial agents detected in our study have also been reported by other authors as frequent contaminants of aquatic environments, mainly due to their resistance to degradation, continuous consumption, and hydrophilicity, which affects their mobility in the aquatic environment [22].

### 2.3. Detection and Prevalence of Antibiotic Resistance Genes (ARGs)

The main concern associated with the presence of antimicrobials in the aquatic environment is their demonstrated impact on the development of antibiotic resistance genes (ARGs) in bacteria, thus impacting the widespread presence of antibiotic-resistant bacteria (ARB) throughout the environmental compartments [32]. In our study, PCR tests were employed, targeting a total of 29 ARGs, out of which nine were detected in the samples examined (Figure 1, Table 3). Of the nine ARGs identified, tetracycline efflux pump-encoding *tetK* was observed in the highest percentage, followed by β-lactamase determinant *blaTEM* and aminoglycoside (streptomycin) phosphotransferase encoding *strA* (Figure 1, Table 3). Also, Sims et al. [24] found aminoglycoside and tetracycline resistance determinants to be the most frequent in their one-year wastewater-based epidemiology study conducted in the city of Bath. Many authors associate high abundance of certain types of ARGs with the close proximity of human or animal wastewater discharge sites, as well as sites where increased concentrations of antimicrobial agents, such as pharmaceutical companies and hospitals, are discharged [22,28,34]. This is also due to the fact that particular types of ARGs dominate the human gut resistome, and these include genes conferring the resistance to tetracyclines, aminoglycosides, β-lactams, macrolide–lincosamide–streptogramins (MLS), and vancomycin [28]. In this study, the areas directly exposed to anthropogenic impacts, in the form of WWTP discharge sites and hospitals, were also characterized by a broader variety of ARGs. For instance, the percentage share of all but one gene that was detected in this study was the highest in the BDZ site (directly impacted by the wastewater discharge and close proximity of a hospital), where eight out of nine detected genes were present (Table 3). The difference in the sum of detected antibiotic resistance genes was significant in the case of BDZ (both water and snow) and R (river water) (F = 4.566, *p* = 0.002; Appendix A_statistical analysis). Both hospital and WWTP effluents have been demonstrated by other authors as significant sources of antibiotic resistance genes in the receiving waters [13,35,36].

Combining analyses of antibiotics presence and concentrations with ARGs can help reveal whether and how the usage of antimicrobials impacts the development of antimicrobial resistance [24]. Several studies have attempted to investigate relationships between the abundance of antimicrobials in the environment and the respective genes with varying outcomes. Li et al. [37] found correlations between the tetracycline levels and the genes determining the resistance to tetracyclines, Rodriguez-Mozaz [38] reported significant correlations between ciprofloxacin and *qnrS*, ofloxacin and *qnrS*, clarithromycin and *ermB*, sulfamethoxazole and *sul1* in wastewater. On the other hand, Sims et al. [24] found no correlation between the antimicrobials and ARGs. In this study, we sought correlations between the prevalence and concentrations of the detected antimicrobials and the prevalence of ARGs in the examined sites (Appendix A_statistical analysis). We found a very strong positive correlation (0.93) between sulfamethoxazole and the *sulIII* gene as well as between oxytetracycline and *blaCTX-M3* (0.98). Erythromycin was correlated with *blaTEM*, *ereA* (erythromycin esterase; 0.71), *strA*, and *sulIII* (Appendix A_statistical analysis). Generally, the *sulIII* gene most frequently correlated with antibiotics in the examined aquatic compartments (i.e., with doxycycline, enrofloxacin, erythromycin, clindamycin, linezolid, ofloxacin, sulfamethoxazole, tetracycline, trimethoprim, vancomycin, and tylosin; Appendix A_statistical analysis). Wilson et al. [39] found the *sulIII* gene in groundwater samples collected at the site where high concentrations of antibiotics were detected, and the distribution of sulfonamide antibiotics coincided with the detection of the *sulIII* gene, suggesting that this gene’s abundance may be related to a higher degree of selection pressure. The *sulIII*, as well as the *tetK* gene (which was the most frequent one in the total pool of samples and the most frequent in the technical snow samples), were among the ones for which more than 10-fold enrichment in wastewater treatment plant effluent compared to influent was observed by Pazda et al. [40]. Pazda et al. [40] explained this phenomenon by the selective pressure that favors bacteria harboring resistance genes and/or horizontal gene transfer between bacteria. Both these genes (i.e., *sulIII* and *tetK*) are located on mobile genetic elements, the presence of which plays an important role in bacterial adaptation to unfavorable conditions and is a means of genetic information transfer among or between bacterial species, thus contributing to their extensive prevalence. On the other hand, Liu et al. [41] found no significant correlation between the ARGs of the *tet* type and the respective tetracycline antibiotics, but the correlation with e.g., sulfonamides was observed. This suggests that the emergence of ARGs may result from the co-selection by antibiotics per se in the environment or other factors, such as temperature and heavy metal concentrations and other environmental pollutants.

However, one of the most important observations from our study is the fact that the prevalence of most ARGs decreases during the technical snow production process (Figure 1). The ski stations examined herein were divided into two groups: the ones where water is stored in technological reservoirs prior to snowmaking and the ones where no such reservoir is in use (Figure 2). It is evident that storage reservoirs contribute to the elimination of ARGs, which are clearly abundant in river water. Among the processes involved in the elimination of ARGs in storage reservoirs, aeration and sedimentation seem the most probable and are also mentioned by other authors as effective ones during ARGs elimination in wastewater treatment facilities [42].

Nevertheless, the *tetK* and *strA* genes can still be found in the samples of technical snow, which suggests that they can be again released into the aquatic environment after snowmelt, but verification of this suspicion requires further studies encompassing the entire cycle of the technical snow production process. Also, among the processes that might contribute to insignificant elimination of ARGs during the technical snowmaking, one can mention the biofilm formation ability by certain bacterial genera, such as, e.g., *Staphylococcus* spp., which can be triggered or enhanced by subinhibitory concentrations of antibiotics [18,43], such as the ones observed in our study. Similarly, as in the case of unexpectedly elevated numbers of bacteria in snow samples, the increased prevalence of ARGs in snowmelt water might result from the detachment of biofilm fragments, including antibiotic-resistant bacteria and resistance determinants. Pei et al. [44], in their study on the ARG response to various biological treatment methods, found that the *tet* and *sul* genes proved more stable at 4 °C than at 20 °C, suggesting that certain conditions may favor the spread and dissemination of ARGs in the environment, and temperature close to freezing may be one such condition. In order to prevent further and unwanted environmental spread of these micropollutants, frequent and thorough maintenance or cleaning procedures should be conducted on the snowmaking devices.

### 2.4. Bacterial Community Diversity and Composition

In order to explore changes in the bacterial community composition during the technical snow production, snowmelt water, reservoir-stored water, and river water samples were analyzed through the V3–V4 16S rRNA Illumina sequencing. The metataxonomic data were used for the assessment of alpha and beta diversity of bacterial communities within the studied sites to determine the bacterial community composition at different taxonomic levels as well as to designate the bacterial genera the prevalence of which differed significantly between the types of samples.

The alpha diversity indices were compared between the samples of water/reservoir/snow and between samples collected in various catchments (Figure 3). Both Shannon and Simpson indices were the highest for river water, and in the case of the Shannon index, the trend was river water > reservoir > technical snow, whereas for the Simpson index it was river water > technical snow > reservoir. In terms of the different catchments, both Shannon and Simpson indices decrease in the following order: R > BDZ > B1 > B3 > BDF, similarly to the decreasing anthropogenic impact put on the examined sites (Table 1). On the other hand, the dominance index, which suggests that one or a few taxa dominate in a community, was the highest in the BDF site and the lowest in the R site when considering the catchments, while the highest was in the reservoir-stored water and the lowest in the technical snowmelt water when considering the river/reservoir/technical snow. These observations suggest that bacterial community diversity may depend on the inflow and availability of growth-supporting substrates, which are presumably higher in wastewater-affected sites and in river or reservoir-stored water than in the pristine sites and technical snow lying on the slopes [45].

Beta-diversity between the bacterial community composition, based on next-generation sequencing of V3–V4 16S rDNA from the water and snow samples from the examined sites, was estimated based on the Bray–Curtis dissimilarity and hierarchical clustering analysis of the samples. The analysis indicated that bacterial communities clustered into three distinct groups (Figure 4), which consist of various types of samples and different sites. The first group clusters are the most pristine sites/samples, i.e., BDF_W, BDF_R, and B1_S; the close second clusters are the two snow samples from anthropogenically impacted sites (R_S and BDZ_S), river water and snowmelt water from the Białka river (B1_W and B3_S), as well as the BDZ_W samples. Finally, the third cluster shows high similarity between the Białka river sites, associating B1_R and B3_S samples. The two remaining samples, i.e., BDF_S and R_W, are the two most distant ones, as also confirmed by the Bray–Curtis dissimilarity values within the heatmap (Figure 4).

The above-described diversity within the considered groups seems to be corroborated by the bacterial community composition analyses at the phylum, family, and genus levels. (Figure 5a–c). At the phylum level (Figure 5a), the bacterial community composition showed significant differences both between the types of samples and the sites examined. *Proteobacteria* and *Bacteroidetes* were the two most prevalent phyla, but their share in the total composition of samples varied clearly. Only in the B_3 site was bacterial community composition dominated by the same phylum (*Proteobacteria*), whereas in all other sites the technical snowmelt water evidently differed from river and reservoir-stored water. In summary, the three most prevalent phyla comprised *Proteobacteria*, *Bacteroidetes*, and *Actinobacteria*, and these are found to be general components of freshwater bacterial communities worldwide [46,47].

At the family level (Figure 5b), *Flavobacteriaceae* was evidently the most prevalent one. It dominated in water samples of the most pristine BDF site, B1_S, as well as in water of the most anthropogenically impacted site R. This is not surprising, as this family comprises environmental genera widely present in freshwater, seawater, and sediments, and only some species are considered potential pathogens of humans and animals [48]. Interestingly, the second most abundant family among the examined samples was *Enterobacteriaceae*, which prevailed in two snow samples, namely the one from the most anthropogenically impacted site and the one from one of the most pristine sites. One of the possible explanations is the fact that members of the *Enterobacteriaceae* family have been found to be able to survive low-temperature marine environments and can persist for a long time in a nonrecoverable but viable state [49]. Thus, even if members of this family constituted a small proportion of a bacterial community (i.e., relative abundance of *Enterobacteriaceae* in R_W is 0.13%), the technical snow production process, which involves freezing at below −4 °C, might have promoted the dominance of these bacteria over others. At B3_W, *Enterobacteriaceae* comprised 19.57%, but the species richness within the B3 site was one of the smallest; hence, the situation might have been similar as in the R site.

Finally, at the genus level, *Deinococcus* clearly dominated in three samples. Interestingly, in the BDF site this genus dominated in the river and reservoir water to sharply drop in the snow, whereas in the B1 site the proportion of this genus changes the other way, i.e., in the river water its proportion is low and increases in the reservoir-stored water to reach the highest levels in snowmelt water (Figure 5c). Species of the genus *Deinococcus* are known for their resistance to multiple stressors, such as ionizing radiation, desiccation, oxidative stress, and high concentrations of xenobiotics like ethyl acetate, toluene, and diethylphtalate, and have been isolated from a wide range of habitats, including oceans, deserts, hot springs, cold polar regions, severely contaminated sites, etc. [50,51]. What needs to be mentioned here is the fact that microorganisms present abundantly in technical snow will inevitably re-enter the environment with meltwater [6]. Thus, an important aspect to consider is the possible impact that these microorganisms may have on, e.g., soil microbiota composition [6,7] and the possible effects on vegetation. For instance, *Deinococcus radiodurans*-derived IrrE protein has proved to significantly enhance salt tolerance in, e.g., *Brassica napus*; thus, the *irrE* gene has been proposed as a potentially promising transgene to improve abiotic stress in crop plants [52]. The second most abundant was the *Escherichia*–*Shigella* group, which dominated in the same two snow samples as the *Enterobacteriaceae* family, so the explanation of its prevalence is probably the same. Nevertheless, water next to manure is considered to be an important transmission vehicle for *E. coli* transfer to plant root zones, and thus its enhanced prevalence in soil, due to the contaminated snowmelt water, may have severe health effects [53]. Importantly, cells of *E. coli* were shown to adhere rapidly to the leaf surface, then showing remarkable resistance to disinfecting agents [54]. *Flavobacterium*, *Rhodococcus*, *Geothrix*, *Truepera*, and *Pseudomonas* were also abundantly present, and—identical to the case of the previously described genera—their share differed clearly between the samples of water and snow (Figure 5c). The relative abundance of both *Flavobacterium* and *Rhodococcus* at BDZ_S was the highest and reached 10% (Figure 5c). Both these genera include species recognized as pathogens; others contribute positively to plant health and development by growth promotion, disease control, tolerance to abiotic stress, degradation of pesticides, siderophore production, and metal scavenging [55,56].

The above-listed genera, as well as several other genera, appeared to be the ones whose abundance was significantly different between the resource water and technical snow (Figure 6). In the case of the B1 site, both river water and reservoir water were included in Figure 6, as the differences between these two samples were evident, unlike in the case of the BDF site. It is clear that the differences in bacterial abundance vary between the examined sites, and these variations include not only different bacterial genera that were enriched or substantially decreased during the technical snow production process. For example, the abundance of *Deinococcus* and *Truepera* increased in the B1 site by more than 400% and more than 1100%, respectively, in reservoir-stored water compared to river water and further increased by more than 4000% and 1800%, respectively, in technical snowmelt water, but it decreased by 96% in the BDF site. One possible reason for this situation may be the fact that the composition of bacterial communities in the aquatic environments is a result of direct pollution from the WWTP effluents, the selective pressure of antimicrobial agents, and the complex ecological interactions among bacteria, which can vary greatly between the examined sites [57].

### 2.5. Multivariate Analysis of Data

As a last step, principal component analysis (PCA) was applied to determine the interrelations between the studied parameters and the factors that may most significantly shape the bacterial community within the examined samples of water and technical snow. The results (Figure 7A,B) indicate that the first three factors explain 59.05% of the variation within the dataset and allow us to identify the most significant variables (Appendix A_PCA variables) determining the resulting differences between the samples. PC1 accounted for 24.67% of variance and pointed to the separation of the more anthropogenically impacted sites (i.e., BDZ and R) from the less impacted ones (B1, B3, and BDF). It also shows the very close position of the least-impacted BDF site (all samples) to the water and snow at B1 site. In this component, five antimicrobial agents—ciprofloxacin, erythromycin, clindamycin, ofloxacin, and trimethoprim—are the most important variables influencing the variations in bacterial community composition (within which *Aurantimicrobium*, *Arthrobacter*, *Atopococcus*, *Carnobacterium*, *Lactococcus*, *Blautia*, and *Acinetobacter* are the most significantly impacted genera) and the sum of antibiotic resistance genes.

Correlation coefficients (Appendix A_statisical analysis) show very similar interrelationships between the concentrations of antimicrobial agents and bacterial community composition, with the same bacterial genera highly correlated with high concentrations of the above-listed antimicrobials. Erythromycin and clindamycin were also positively correlated with four antibiotic resistance determinants, i.e., *blaTEM*, *ereA*, *strA*, and *sulIII*, while ofloxacin and trimethoprim—with *sulIII*. This indicates the significant impact antimicrobial agents have not only on antimicrobial resistance determinants in the environment but also on bacterial community composition in the aquatic environment, which has also been mentioned by other authors [58,59,60].

PC2, which explained 19.91% of variance among the samples, points to the most significant differences between technical snow at B3 and R sites and BDZ_W. In this component only the NGS reads had the highest loadings, and these were *Legionellaceae*, *Neisseriaceae*, *Amycolatopsis*, *Propionibacterium*, *Micrococaceae*, followed by *Bacillaceae*, *Sphingobacterium*, *Enterobacteriaceae*, *Staphylococcus*, and *Escherichia*. Finally, the third component (PC3) explained 14.47% of variance between the samples and this factor was positively correlated to culturable bacteria, i.e., *E. coli*, *Staphylococcus* and *Salmonella*, antibiotic vancomycin, and NGS-based reads of *Pseudomonas*, while negatively with culture-based enterococci (*E. faecalis*/*E. faecium*), antibiotic ciprofloxacin, NGS-based *Carnobacterium*, and *Hypnocyclicus* (Appendix A).

Correlation analysis also showed positive relations between vancomycin concentrations in the samples and culture-based results of *E. coli*, *Staphylococcus*, and *Salmonella*, as well as *Rhodococcus*, *Glutamicibacter*, *Pseudarthrobacter*, and *Pseudomonas* (Appendix A). It also most visibly demonstrates the diversity between the water and technical snow of one of the most anthropogenically impacted sites (BDZ). Also, the water and snow samples of the second anthropogenically impacted site (R) are located on the opposite sides of the axis but much closer to one another (Figure 7B). This factor points clearly to the changes resulting from the technical snow production process and affecting both antibiotic concentrations in the samples and the bacterial community composition, as both concentrations of vancomycin and ciprofloxacin change during the technical snow production at BDZ and R sites (Table 1) as well as the abundance of *Carnobacterium*, *Rhodococcus*, *Pseudarthrobacter*, and *Pseudomonas* (Figure 6). One of the above listed, highly correlated with vancomycin, genera—*Glutamicibacter*, even though typically found in a variety of environments and generally considered non-harmful, has been mentioned as a pathogen associated with urinary tract infections and bacteremia [61]. On the other hand, *Pseudarthrobacter*, the abundance of which decreased in technical snow of the BDZ site compared to river water, has been considered a plant growth-promoting bacterium, able to increase, e.g., flavonoid content of plants, showing high IAA content, nitrogen-fixing, potassium- and phosphate-solubilizing properties, and positively affecting the growth of various plants [62,63]. Its closely related genus, *Arthrobacter*, acting as a tool for bioremediation in agriculture and a plant growth promoter due to e.g., nitrogen fixation, potassium and phosphate solubilization, and indole acetic acid synthesis [63], was positively correlated with the antibiotics ciprofloxacin, ofloxacin, and clindamycin, and yet its abundance increased at BDZ_S compared to BDZ_W. These observations suggest that there are still many complex interrelations between the biotic and abiotic components of the aquatic environment to be understood and studied.

## 3. Materials and Methods

### 3.1. Sampling Sites

The study sites comprised five ski stations where technical snow is produced from nearby rivers and streams. The ski stations are situated in the catchments of three rivers in southern Poland: Białka (two stations: B1 and B3), Biały Dunajec (two stations: BDF and BDZ), and Raba (R). Two out of the five ski stations (B1 and BDF) collect water in technical reservoirs prior to and during the winter seasons. Water is supplied to the reservoirs from the rivers and streams located in the direct vicinity of the ski stations, while the remaining three stations (B3, BDZ, and R) produce technical snow withdrawn directly from the rivers and streams via the pipelines that supply water to snow cannons. The precise location of the ski stations studied cannot be disclosed due to the confidentiality agreements with the ski station companies, but their area is shown in Appendix A. The five ski stations examined vary in terms of the location of their water intakes for the technical snow production systems relative to the points of river water pollution. This translates into the anthropogenic pressure put on the water resources in the direct vicinity of the ski stations, which can be arranged in the following ascending order: BDF > B3 > B1 > R > BDZ (Table 4).

**Table 4 ijms-26-02771-t004:** Description of the study sites. The color intensity of table rows refers to the level of anthropogenic pressure, i.e. first row—the smallest anthropopressure, the last row—the highest anhropopressure.

Code	Height Above Sea Level [m a.s.l]	Number of Inhabitants	Anthropogenic Pressure Description	Technical Reservoir [yes/no]	Sample Description and Code
BDF	850	540	upstream of a small village next to the Tatra National Park, upstream of wastewater discharge sites	yes	river water (BDF_W)
					storage reservoir (BDF_R)
					snowmelt water (BDF_S)
B3	760	950	upstream of a small village next to the Tatra National Park and the Polish/Slovakian border	no	river water (B3_W)
					snowmelt water (B3_S)
B1	700	2300	center of a popular tourist resort, ca. 3 km downstream of a WWTP	yes	river water (B1_W)
					storage reservoir (B1_R)
					snowmelt water (B1_S)
R	315	17,500	center of a medium-sized town, downstream of several ski resorts, ca. 5 km downstream of a hospital, ca. 10 km of a WWTP	no	river water (R_W)
					snowmelt water (R_S)
BDZ	750	25,000	center of a popular tourist resort, ca. 3 km downstream of a WWTP, ca. 2 km downstream of a hospital	no	river water (BDZ_W) snowmelt water (BDZ_S)

### 3.2. Sample Collection

The following samples were collected: river water at intakes for technical snow production (five sites), technical reservoirs (two sites), and freshly produced technical snow (five sites). The samples were collected in two or three campaigns (depending on the weather conditions, i.e., when the ambient temperature was low enough (i.e., below −4 °C) to allow for the technical snow production) between late November and early January. In total, 31 individual samples were examined, comprising n = 13 river water, n = 4 technical reservoir, and n = 14 melt water from freshly produced technical snow.

During sampling, the samples were collected in three instantaneous replications that formed the final mixed sample. River and reservoir-stored water was collected into sets of 1000 mL sterile polypropylene bottles while snow was collected by first scratching the superficial layer, followed by the collection of snow with a 1.0 m long, 10 cm wide snow corer and transferring the snow into sterile plastic bags where it was stored until melting. Then, snowmelt water was poured into sets of sterile 1000 mL polypropylene bottles and analyzed in the same way as river and reservoir water.

### 3.3. Culture-Based Microbiological Analysis of Samples

The membrane filtration method was used to enumerate *Escherichia coli* and *Enterococcus faecalis*/*E. faecium* in 100 mL of water, and the pour plate method was used to enumerate coagulase-positive *Staphylococcus* (including *S. aureus*) and *Salmonella* in 1 mL of water. *E. coli* was grown on Tryptone Bile-X-glucuronide agar (TBX) agar (Biomaxima, Lublin, Poland) and incubated at 44 °C for 24 h (blue-green colonies were counted as *E. coli*), *E. faecalis*/*E. faecium* were grown on Slanetz–Bartley agar (Biomaxima, Lublin, Poland) (dark red to light brown colonies after incubation at 37 °C for 72 h), *Salmonella* were grown on SS agar (Biomaxima, Lublin, Poland) (black colonies after incubation at 37 °C for 24 h), and coagulase-positive staphylococci were grown on Baird–Parker agar (dark grey to black colonies with a halo after incubation at 37 °C for 48 h). After incubation, visible and characteristic colonies were counted, and the results were expressed as the number of colony-forming units per 100 mL (CFU/100 mL) or per 1 mL (CFU/mL).

### 3.4. Determining the Presence and Concentration of Antimicrobial Agents in Water and Snowmelt Samples

Antimicrobial agents were selected for the analysis based on their wide application in human and veterinary medicine in Europe, which was determined based on documents such as reports from the European Centre for Disease Prevention and Control [64] and the European Medicines Agency [65]. In total, 21 antimicrobial agents were examined, which belonged to 14 classes: aminopenicillins (amoxicillin and ampicillin), 2nd gen. cephalosporins (cefoxitin), 3rd gen. cephalosporins (ceftazidime), fluoroquinolones (ciprofloxacin, enrofloxacin and ofloxacin), lincosamides (clindamycin), tetracyclines (doxycycline, oxytetracycline and tetracycline), macrolides (erythromycin and tylosin), aminoglycosides (gentamycin and netilmycin), oxazolidinones (linezolid), carbapenems (meropenem), semi-synthetic penicillins (piperacillin), sulphonamids (sulfamethoxazole), dihydrofolate reductase inhibitors (trimethoprim), and glycopeptides (vancomycin).

Antimicrobial agents were extracted from the water samples by solid-phase extraction (SPE) using Oasis HLB cartridges (6 cc Vac 500 mg Sorbent per cartridge, 60 μm Particle Size, Waters, Milford, CT, USA), according to the procedure described in detail by [66]. The quantitative assessment of antibiotic concentration in the examined samples was conducted using an ultra-high-performance liquid chromatography (UHPLC) device equipped with an autosampler (Agilent 1290 Infinity System) and mass spectrometer (MS) Agilent 6460 Triple Quad Detector (Santa Clara, CA, USA) [66]. The presence of antimicrobials in the samples was verified based on their product ions resulting from the decay of precursor ions, developed by Lenart-Boroń et al. [5]. The limits of detection (LOD) ranged from 0.083 to 83.3 ng/L, and the limits of quantification ranged from 0.25 to 250 ng/L, while the SPE recovery ranged from 9.94% to ca. 100% for various antimicrobial agents [66].

### 3.5. PCR Determination of Genetic Antimicrobial Resistance Determinants in Total Genomic DNA

The presence of 29 genes encoding various mechanisms of bacterial resistance to antimicrobial agents was assessed in total DNA extracts from river water, reservoir-stored water, and snowmelt water. The following gene classes were examined (Table 4): extended-spectrum beta lactamase determinants: *blaTEM*, *blaCTX-M*, *blaCTX-M3*, *blaCTX-M9*, *blaSHV*, *blaOXA-1*, *blaOXA-48*, *blaKPC*, *blaIMP*, *blaVIM*, and *blaNDM*; methicillin-resistance determinant *mecA*; macrolide–lincosamide–streptogramin B resistance encoding genes: *ereA*, *ereB*, *ermA*, *ermB*, *msrA*, *msrB*, *mphA*, *lnuA*, *vatA*, *vatB*, *vga*, and *vgb*; and others: *strA*, *dfrA12*, *aac6*, *tetK*, and *sulIII*.

Total bacterial DNA was extracted from 200 mL of water filtrates, collected on 0.22 µm sterile cellulose nitrate filters. After filtration, 1 mL of sterile 0.85% NaCl solution was poured on filters and stirred overnight at 120 rpm. Then, 1 mL of supernatant was collected, and the filter surfaces were thoroughly swabbed, and the entire content was transferred to 1.5 mL Eppendorf tubes. DNA was extracted using a Genomic Mini AX Bacteria+ Spin extraction kit (A&A Biotechnology, Gdańsk, Poland), following the manufacturer’s instructions.

PCR tests were performed in a 25 µL volume each, containing ca. 50 ng of DNA template, 12.5 pM of each primer, 2.0 mM of dNTP, 1 × PCR buffer, and 2.4 U Taq DNA polymerase (PCR Mix Plus Green, A&A Biotechnology, Gdańsk, Poland). The reactions were performed in a T100 thermal cycler (BioRad, Hercules, CA, USA), under temperature conditions optimal for the respective primers (Table 5). The resulting products were visualized in UV light after electrophoresis in 1% agarose gel in 1 × TBE buffer, stained with SimplySafe (EurX, Gdańsk, Poland), with a DNA 3 size marker (A&A Biotechnology, Gdańsk, Poland).

### 3.6. Illumina Sequencing of V3-V4 16S rRNA Amplicon

All samples in a total volume of 500 mL were vacuum filtered through 0.22 µm sterile cellulose nitrate filters (Sartorius, Göttingen, Germany), and bacterial genomic DNA was extracted using a Genomic Mini AX Bacteria + extraction kit (A&A Biotechnology, Gdańsk, Poland), followed by DNA purification using the Anti-Inhibitor Kit (A&A Biotechnology, Gdańsk, Poland). DNA concentration was measured fluorometrically on a Qbit 4 Fluorometer (ThermoFisher Scientific, Waltham, WA, USA). Amplicon libraries of the hypervariable V3-V4 region of the 16S rRNA gene were prepared according to the 16S Metagenomic Sequencing Library Preparation Part # 15044223 Rev. B (Illumina, San Diego, CA, USA), followed by a two-step PCR using the 16S amplicon PCR primer Forward (5′-TCGTCGGCAGCGTCAGATGTGTATAAGAGACAGCCTACGGGNGGCWGCAG-3′) and the 16S Amplicon PCR Reverse Primer (5′-GTCTCGTGGGCTCGGAGATGTGTATAAGAGACAGGACTACHVGGGTATCTAATCC-3′) [80] and Herculase II Fusion DNA Polymerase Nextera XT Index Kit V2 (Agilent Technologies, Santa Clara, CA, USA). The sample libraries were loaded on an Illumina MiSeq platform, and 2 × 300 bp reads were generated by Macrogen (Seoul, South Korea).

### 3.7. Statistical Analyses

The normality of data was assessed using the Shapiro–Wilk’s test. As the data distribution was close to normal, parametric tests were used in further analyses. Analysis of variance followed by the post hoc least significant difference test was used to assess the significance of differences in the numbers of microorganisms, the concentrations of antimicrobial agents, and the sum of genes determining antibiotic resistance between the examined sites. Pearson’s correlation coefficient was used to assess the correlation between the following parameters: numbers of culturable bacteria, antibiotic concentrations, the sum of genes determining antimicrobial resistance, and numbers of reads in identified genera (Illumina). Principal component analysis (PCA) was applied to explore the relationship between numbers of microorganisms, antibiotic concentrations, the sum of genes determining antimicrobial resistance, and numbers of reads in identified genera (Illumina). The number of principal components and the factors were selected according to the Kaiser criterion, and the factors with eigenvalues >1.00 were taken into consideration. The tests were conducted in Statistica v. 13 (TIBCO Software, Palo Alto, Santa Clara, CA, USA).

The 16S rRNA V3–V4 regions from the Illumina sequencing were identified by comparing the sequence reads against the Greengenes v.13 database (97% similarity, minimum score 40). The resulting sequences in the form of operational taxonomic units (OTUs) were taxonomically assigned to the phylum level or lower ranks using CLC Genomics Workbench v. 13 (Qiagen, Hilden, Germany) and Microbial Genomics Module Plugin v. 4.1 (Qiagen, Hilden, Germany). Data obtained on the occurrence of bacterial taxa were used to calculate the relative abundance of the most common bacterial genera and phyla. The calculations were conducted, and graphs were constructed in R software (version 4.4.2) using the following packages: vegan [81], ggplot2 [82], and pheatmap [83].

## 4. Conclusions

In this study, we examined whether and how the technical snow production from water resources of varying bacterial community composition, antibiotic resistance genes, and antimicrobial agents content alters these parameters that may, in turn, affect the overall condition of the surrounding environment and health of people that come into direct contact with the technical snow. With the dearth of studies conducted regarding technical snowmaking and the increasing importance of technical snowmaking for tourism around the world, this study attempted to indicate the most important threats that snowmaking poses to human and environmental health, as well as to propose possible solutions for mitigating the identified risks.

The most important findings of our study are as follows:○The numbers of culturable bacteria drop sharply during the technical snowmaking process; thus, even if the snow is produced from water severely contaminated by wastewater, the resulting snow does not seem to pose a serious threat to human health.○The presence and concentration of antimicrobial agents in water and the produced snow is strongly affected by the proximity of point sources of pollution, such as WWTPs and hospitals.○As many as nine antibiotic resistance genes (ARGs) were detected at the examined sites, and their prevalence was strongly affected by the close vicinity of wastewater discharge sites and hospitals. The presence and concentration of antimicrobial agents is also associated with the occurrence of ARGs.○Importantly, the prevalence of most ARGs decreased during technical snowmaking. This was more evident within ski stations where water is stored in technical reservoirs prior to snowmaking. On the other hand, biofilm formation and its further detachment may contribute to reduced removal efficiency of ARGs.○Finally, the NGS-based analysis of bacterial community composition evidently indicates its changes through the process of technical snowmaking, which may be due to the survivability of certain strains in freezing temperatures or the inhibitory effect of antimicrobial agents that enter the technical snow.○The presence and concentration of antimicrobials in water seem to be the most significant factors affecting the changes in bacterial community composition. Therefore, measures should be taken to reduce their spread within the aquatic environment.○Among the measures that a ski station management could implement, one can mention the construction of technical reservoirs. These appear to effectively eliminate pollutants and micropollutants from the resource water. In addition, ensuring the regular and frequent cleaning and maintenance of snowmaking devices is advisable.

Despite concerns about the impact it has on the surrounding environment or health outcomes it may have if technical snow is made of contaminated water or reclaimed water, the percentage of ski areas that use or rely on technical snow has gradually increased, and as such, this trend is not likely to reverse. Therefore, research is needed to address the still not elucidated issues of, for example, the impact that melting technical snow can have on the receiving waters and soil. This can be examined in terms of bacteriological contamination, antibiotic content, and antibiotic resistance genes, as well as in-depth, sequencing-based assessment of alterations of local bacterial community compositions.

## Figures and Tables

**Figure 1 ijms-26-02771-f001:**
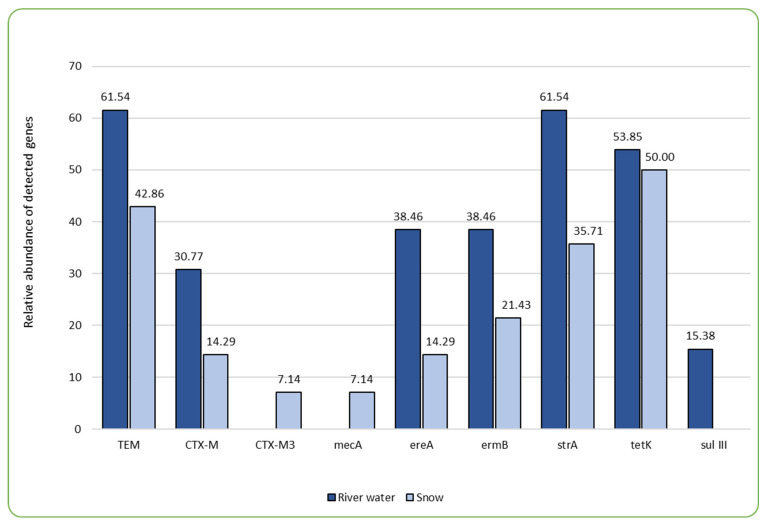
The comparison of relative abundance (%) of genes determining antibiotic resistance present in the samples of river water and technical snow.

**Figure 2 ijms-26-02771-f002:**
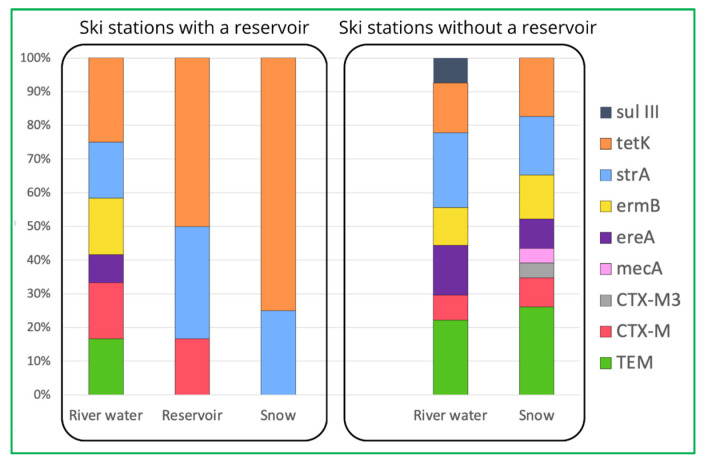
Comparison of prevalence of detected ARGs in ski stations with and without reservoirs.

**Figure 3 ijms-26-02771-f003:**
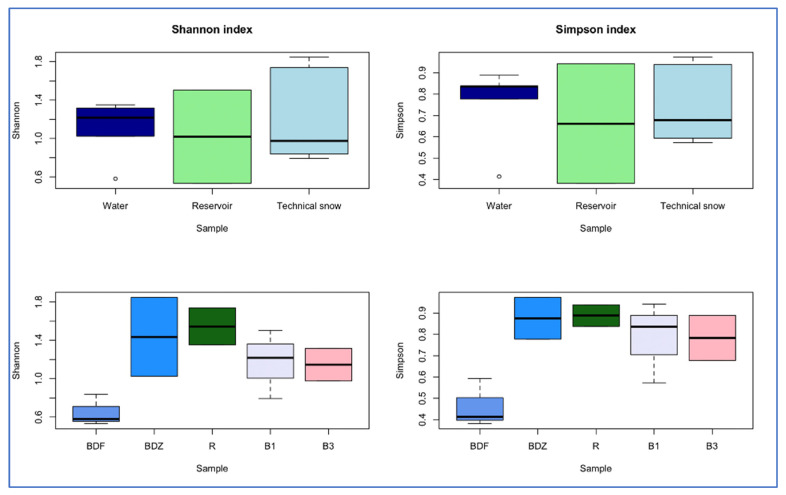
Differences in alpha-diversity indices between the types of samples (river water, reservoir water, and technical snow) and between the examined sites.

**Figure 4 ijms-26-02771-f004:**
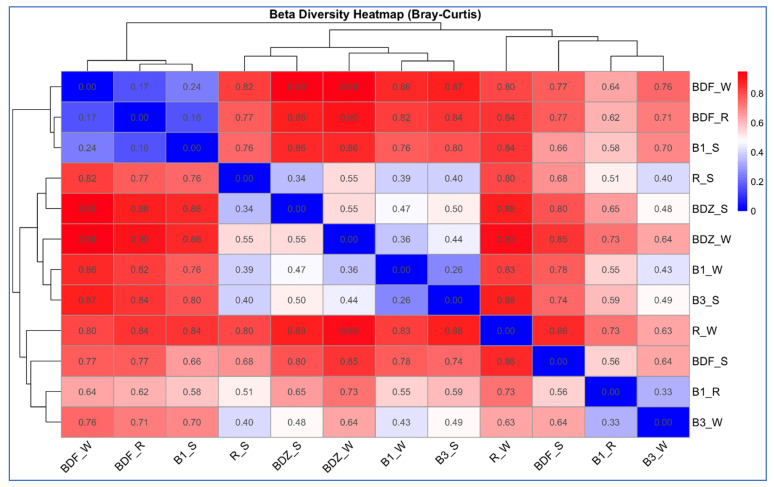
Hierarchical clustering and a heatmap of pairwise Bray–Curtis dissimilarity between bacterial communities of individual samples (0 indicates identical community composition and 1 indicates completely different compositions of compared samples).

**Figure 5 ijms-26-02771-f005:**
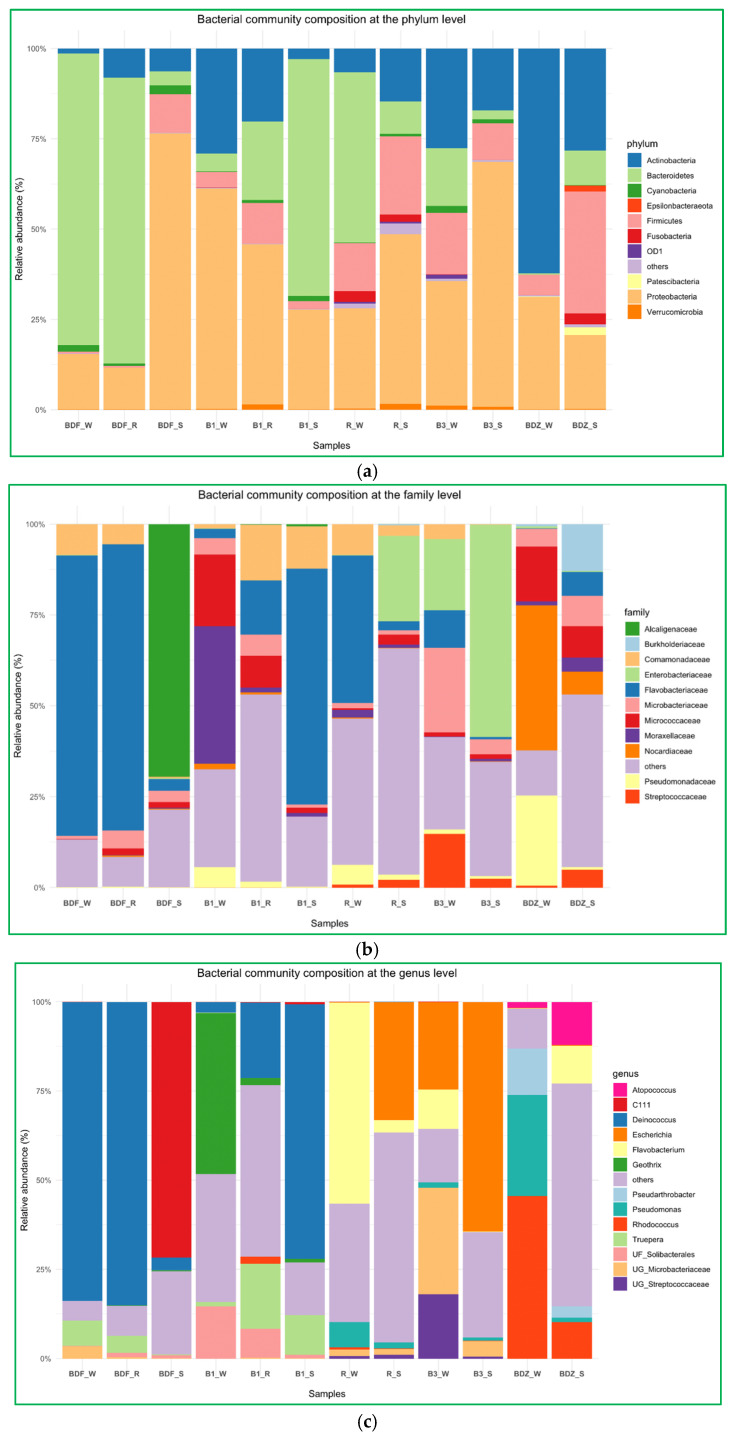
Bacterial community composition of river water, reservoir water, and snowmelt water samples, (**a**) at the phylum level, (**b**) at the family level, and (**c**) at the genus level.

**Figure 6 ijms-26-02771-f006:**
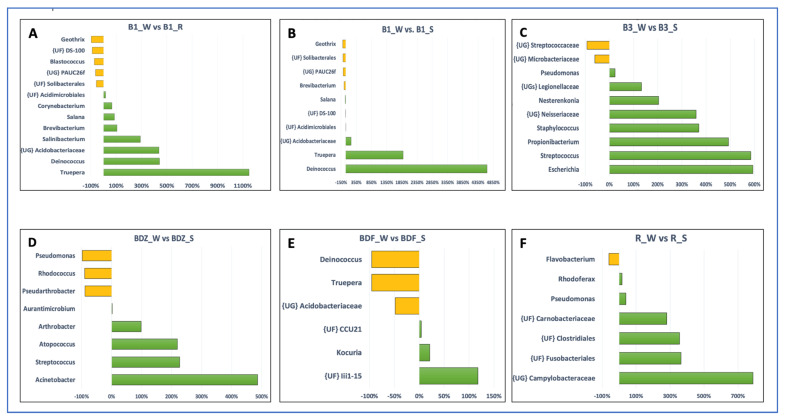
Bar plots depicting the significant differences in the number of reads between the water and snow or reservoir water samples at the genus level (analysis was conducted for OTUs exceeding the threshold of 100 reads). Letters (**A**–**F**) refer to individual sites ((**A**,**B**)—B1 site; (**C**)—B3 site; (**D**)—BDZ; (**E**)—BDF; (**F**)—R site). The green bars represent taxa the numbers of which increased and the yellow bars represent taxa the numbers of which decrease between the two types of samples (i.e. river water vs. reservoir water or river water vs. snowmelt water).

**Figure 7 ijms-26-02771-f007:**
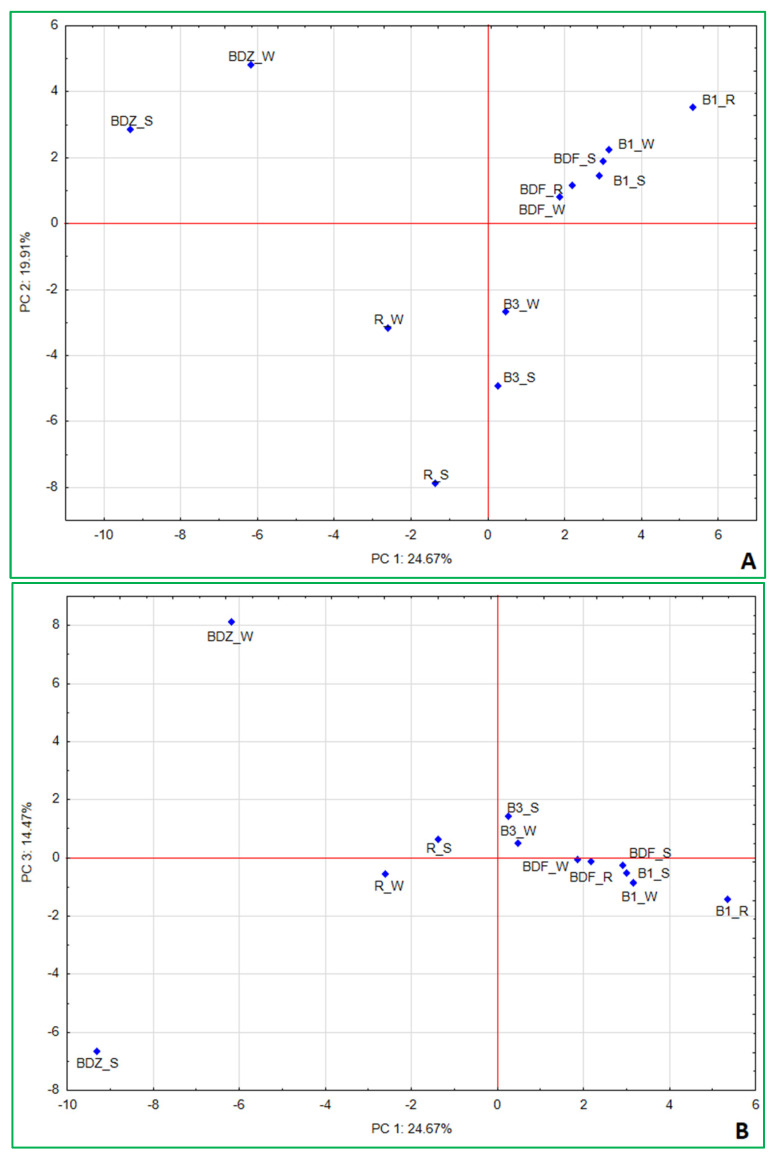
Principal component analysis (PCA) based on the bacterial community composition at the genus level, culturable bacteria, concentrations of antibiotics, and sum of genetic determinants of antibiotic resistance in a given sample. (**A**) PC1/PC2; (**B**) PC1/PC3.

**Table 1 ijms-26-02771-t001:** Mean numbers of bacterial indicators of water quality (CFU/mL and CFU/100 mL) at the study sites *. The intensity of the background color of table rows refers to the intensity of anthropogenic pressure and the resulting bacteriological contamination of the sites, i.e. the smallest anthropopressure and bacteriological contamination was in sample BDF, while the highest–in sample BDZ.

Code	Sample Description and Code	*E. coli*	*E. faecalis*/*E. faecium*	*Salmonella*	Coagulase-Positive *Staphylococci*
		CFU/100 mL		CFU/mL	
BDF	river water (BDF_W)	1	4	1	0
	storage reservoir (BDF_R)	0	5	0	350
	snowmelt water (BDF_S)	1	7	1	0
B3	river water (B3_W)	119	45	0	2
	snowmelt water (B3_S)	0	26	0	2
B1	river water (B1_W)	186	94	0	0
	storage reservoir (B1_R)	14	13	0	0
	snowmelt water (B1_S)	2	7	0	12
R	river water (R_W)	298	45	56	328
	snowmelt water (R_S)	87	26	5	232
BDZ	river water (BDZ_W)	224,067	273,641	10,785	8667
	snowmelt water (BDZ_S)	153	286	4	190

* indicates site descriptions with code explanations are shown in Table 4 in Materials and Methods section.

**Table 2 ijms-26-02771-t002:** Mean concentrations of antimicrobial agents at the examined sites * (the values are presented as means of all measurements); the overall frequency of antimicrobial agents’ detection in all samples (the values are presented as a percentage of detection in all examined samples, *n* = 31); and the total concentrations of antimicrobial agents (presented as the sum of all measurements within the study).

Chemical Group	Antibiotic	BDF_W	BDF_R	BDF_S	B1_W	B1_R	B1_S	R_W	R_S	B3_W	B3_S	BDZ_W	BDZ_S	Frequency of Detection (% of All Samples)	Total Concentration of Antibiotics in All Samples (ng/L)
2nd gen. cephalosporins	cefoxitin	0.00	0.00	112.59	0.00	0.00	73.47	0.00	0.00	0.00	0.00	0.00	0.00	9.7	519.05
fluoroquinolones	ciprofloxacin	0.00	0.00	0.00	0.00	0.00	0.00	65.20	24.31	0.00	0.00	0.00	15.37	9.7	314.65
	enrofloxacin	0.00	0.00	0.00	1.67	6.09	3.04	626.28	34.95	3.38	0.95	2.94	0.00	3.2	204.78
	ofloxacin	0.00	0.00	0.00	0.49	0.00	0.00	95.64	28.39	0.00	0.00	7.00	6.29	29.0	2030.53
lincosamids	clindamycin	0.51	0.00	0.00	1.83	<LOQ **	2.47	37.66	9.96	5.80	0.00	15.59	11.18	29.0	3.71
macrolides	erythromycin	0.10	0.00	0.00	0.07	0.00	0.05	0.30	0.24	0.00	0.00	0.30	0.20	71.0	251.15
	tylosin	0.00	0.00	0.00	0.00	0.00	0.00	56.59	10.23	0.00	0.00	0.00	0.00	22.6	64.45
tetracyclines	doxycycline	0.00	0.00	0.00	0.00	0.00	0.00	68.26	0.00	0.00	0.00	0.00	0.00	16.1	413.43
	oxytetracycline	0.00	0.00	0.00	0.00	0.00	0.00	13.47	0.00	0.00	0.00	0.00	61.36	6.5	224.46
	tetracycline	0.00	0.00	0.00	0.00	0.00	0.00	9.23	0.00	0.00	0.00	0.00	0.00	35.5	214.61
sulphonamids	sulfamethoxazole	0.00	0.00	0.00	0.00	0.00	1.56	20.90	4.69	1.48	0.00	34.05	8.83	3.2	27.70
antifolates	trimethoprim	0.00	0.00	0.00	1.10	0.00	0.18	38.57	7.70	2.59	0.00	8.61	4.93	35.5	188.63
glycopeptides	vancomycin	0.00	0.00	0.00	0.00	0.00	0.00	142.66	21.84	0.00	0.00	4.23	0.00	9.7	506.19
oxazolidinones	linezolid	0.00	0.00	0.00	0.00	0.00	1.33	17.40	1.48	1.63	0.00	3.99	1.55	6.5	200.44
number of antibiotics detected		2	0	1	5	2	7	13	10	5	1	8	8		
total concentration of antibiotics		1.21	0	225.19	15.45	12.17	328.40	3559.06	431.37	29.77	1.90	230.14	329.10		

* indicates codes of sampling sites are described in the Section 3 describing sampling procedures, in Table 4. ** LOQ—limit of quantification

**Table 3 ijms-26-02771-t003:** Distribution of the detected antibiotic resistance genes in the examined ski stations (site abbreviations are provided in Section 3, Table 4).

Site	Beta-Lactamases			Altered Penicillin-Binding Protein (PBP2a)	Erythromycin Esterase	Macrolide Ribosomal Methylase	Aminoglycoside 3’-phosphotransferase	Tetracycline Efflux Protein	Dihydropteroate Synthase
	*blaTEM*	*blaCTX-M*	*blaCTX-M3*	*mecA*	*ereA*	*ermB*	*strA*	*tetK*	*sulIII*
BDF	16.7	16.7	0	0	0	0	16.7	50.0	0
B3	50.0	25.0	0	0	0	0	0	50.0	0
B1	12.5	12.5	0	0	0	12.5	37.5	75	0
R	66.7	16.7	0	16.7	16.7	0	66.7	33.3	16.7
BDZ	100	33.3	16.7	0	83.3	100	100	66.7	16.7
total share	46.7	20.0	3.3	3.3	20.0	23.3	46.7	56.7	6.7

**Table 5 ijms-26-02771-t005:** PCR primers for antimicrobial resistance genes with primer annealing temperature and product length.

No.	Resistance Mechanism	Gene	Primer	Sequence (5′-3′)	Annealing Temp. (°C)	Product Length (bp)	Reference
1.	Extended-spectrum beta-lactamases (ESBL)	*blaTEM*	blaTEM-F	ATTCTTGAAGACGAAAGGGC ACGCTCAGTGGAACGAAAAC	60	1150	[67]
			blaTEM-R				
2.		*blaCTX-M*	blaCTX-M-F	CGATGTGCAGTACCAGTAA TTAGTGACCAGAATCAGCGG	55	585	[68]
			blaCTX-M-R				
3.		*blaCTX-M3*	blaCTX-M3-F	GTTACAATGTGTGAGAAGCAG CCGTTTCCGCTATTACAAAC	50	1049	[69]
			blaCTX-M3-R				
4.		*blaCTX-M9*	blaCTX-M9-F	GTGACAAAGAGAGTGCAACGG ATGATTCTCGCCGCTGAAGCC	54	856	[70]
			blaCTX-M9-R				
5.		*blaSHV*	blaSHV-F	CACTCAAGGATGTATTGTG TTAGCGTTGCCAGTGCTCG	52	885	[67]
			blaSHV-R				
6.		*blaOXA-1*	blaOXA-1-F	ACACAATACATATCAACTTCGC AGTGTGTTTAGAATGGTGATC	61	813	[67]
			blaOXA-1-R				
7.	Carbapenemases class D	*blaOXA-48*	blaOXA-48-F	GCTTGATCGCCCTCGATT GATTTGCTCCGTGGCCGAAA	60	281	[71]
			blaOXA-48-R				
8.	Carbapenemases class A	*blaKPC*	blaKPC-F	TGTTGCTGAAGGAGTTGGGC ACGACGGCATAGTCATTTGC	57	340	[71]
			blaKPC-R				
9.	Carbapenemases class B	*blaIMP*	blaIMP-F	TTGACACTCCATTTACAG GATCGAGAATTAAGCCACCC	56	139	[71]
			blaIMP-R				
10.		*blaVIM*	blaVIM-F	GATGGTGTTTGGTCGCATA CGAATGCGCAGCACCAG	60	390	[71]
			blaVIM-R				
11.		*blaNDM*	blaNDM-F	GGTTTGGCGATCTGGTTTTC CGGAATGGCTCATCACGATC	60	621	[72]
			blaNDM-R				
12.	Methicillin-resistance	*mecA*	mecA-F	GTAGAAAATGACTGAACGTCCGATAA CCAATTCCACATTGTTTCGGTCTAA	55	310	[73]
			mecA-R				
13.	Macrolide–lincosamide–streptogramin B (MLSb) resistance genes	*ereA*	ereA-F ereA-R	AACACCCTGAACCCAAGGGACG CTTCACATCCGGATTCGCTCGA	57	420	[74]
14.		*ereB*	ereB-F ereB-R	AGAAATGGAGGTTCATACTTACCA CATATAATCATCACCAATGGCA	52	546	[74]
15.		*ermA*	ermA-F ermA-R	TCTAAAAAGCATGTAAAAGAA CTTCGATAGTTTATTAATATTAGT	52	645	[74]
16.		*ermB*	ermB-F ermB-R	GAAAAGGTACTCAACCAAATA AGTAACGGTACTTAAATTGTTTAC	55	639	[74]
17.		*msrA*	msrA-F msrA-R	GGCACAATAAGAGTGTTTAAAGG AAGTTATATCATGAATAGATTGTCCTGTT	50	940	[75]
18.		*msrB*	msrB-F msrB-R	TATGATATCCATAATAATTATCCAATC AAGTTATATCATGAATAGATTGTCCTGTT	50	595	[75]
19.		*mphA*	mphA-F mphA-R	AACTGTACGCACTTGC GGTACTCTTCGTTACC	50	837	[74]
20.		*lnuA*	lnuA-F lnuA-R	GGTGGCTGGGGGGTAGATGTATTAACTGG GCTTCTTTTGAAATACATGGTATTTTTCGATC	57	323	[75]
21.		*vatA*	vatA-F vatA-R	CAATGACCATGGACCTGATC CTTCAGCATTTCGATATCTC	52	619	[75]
22.		*vatB*	vatB-F vatB -R	CCCTGATCCAAATAGCATATATCC CTAAATCAGAGCTACAAAGT	52	602	[75]
23.		*vga*	vga-F vga-R	CCAGAACTGCTATTAGCAGATGAA AAGTTCGTTTCTCTTTTCGACG	54	470	[75]
24.		*vgb*	vgb-F vgb-R	ACTAACCAAGATACAGGACC TTATTGCTTGTCAGCCTTCC	53	734	[75]
25.	*Streptomycin resistance*	*strA*	strA-F strA-R	TCAATCCCGACTTCTTACCG CACCATGGCAAACAACCATA	52	126	[76]
26.	*Trimetophrim resistance*	*dfrA12*	dfrA12-F dfrA12-R	TTTATCTCGTTGCTGCGATG AGGCTTGCCGATAGACTCAA	50	155	[77]
27.	*Aminoglycoside resistance*	aac(6′)/aph(2′)	aac(6′)/aph(2′′)-F aac(6′)/aph(2′′)-R	CAGAGCCTTGGGAAGATGAAG CCTCGTGTAATTCATGTTCTGGC	55	348	[78]
28.	*Tetracyclines resistance*	*tetK*	tetK-F tetK-R	TCGATAGGAACAGCAGTA CAGCAGATCCTACTCCTT	55	169	[79]
29.	*Sulfonamides* *resistance*	*sulIII*	sulIII-F sulIII-R	ACCACCGATAGTTTTTCCGA TGCCTTTTTCTTTTAAAGCC	62	199	[77]

## Data Availability

The data presented in this study are available from the corresponding author upon request.

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
