# Peer review of "Impact of the Technical Snow Production Process on Bacterial Community Composition, Antibacterial Resistance Genes, and Antibiotic Input—A Dual Effect of the Inevitable"

_ijms, 2025, doi:10.3390/ijms26062771_

Round 1
Reviewer 1 Report
Comments and Suggestions for Authors
This article did some research on the bacteria behavior in technical snow production. This topic is attracting and has potential application. However, some issues should be addressed and improved:
- The overall writing format for the paper needs improving. That include the details in the table design.
- Please carefully check the writing of the numbers, especially the scientific notation.
- The discussion part is recommended to rearrange into more paragraphs more easier reading.
- Is there any study on the biofilm formation or bacteria growth rate regarding the technical snow production? Maybe those content could be mentioned in the manuscript.
Author Response
- The overall writing format for the paper needs improving. That include the details in the table design.
Response: the manuscript has been thoroughly corrected and improved as suggested. The Abstract has been rewritten, Introduction and aims of the study were supplemented with some additional information referring to the impact of technical snowmaking on human and environmental health. Unnecessary information concerning the novel techniques was removed. Aims were rephrased and specified. Methods have been supplemented with additional information about statistical analysis. Discussion has been expanded, while detailed description of the results has been shortened or replaced with reference to the corresponding Tables and/or Figures. Conclusions section was shortened and the most important findings have been pointed. The tables were reformatted, i.e. Table 1 was split into two separate tables, additional information to tables has been provided as table captions.
- Please carefully check the writing of the numbers, especially the scientific notation.
Response: The numbers and units have been corrected. The Figure 1 which has been previously pasted as an Excel graph, has now been pasted as a figure to prevent the separators from being changed. However, the original version of this figure has been provided in a file that contains all figures.
- The discussion part is recommended to rearrange into more paragraphs more easier reading.
Response: The discussion has been rearranged as suggested. The long fragments of text have been divided into shorter paragraphs and an additional sub-chapter (describing multivariate statistical analysis) has been added.
- Is there any study on the biofilm formation or bacteria growth rate regarding the technical snow production? Maybe those content could be mentioned in the manuscript.
Response: Yes, we have found a few studies referring to the possibility of bacterial biofilm formation and bacterial growth in the technical snow and cold regions. We have added a few references and included them in Results and Discussion:
a) Faria, C.; Vaz-Moreira, I.; Serapicos, E.; Nunes, O.C.; Manaia, C.M. Antibiotic Resistance in Coagulase Negative Staphylococci Isolated from Wastewater and Drinking Water. Science of the Total Environment 2009, 407, 3876–3882, doi:10.1016/j.scitotenv.2009.02.034.
b) Williams, M.M.; Domingo, J.W.S.; Meckes, M.C.; Kelty, C.A.; Rochon, H.S. Phylogenetic Diversity of Drinking Water Bacteria in a Distribution System Journal of applied microbiology 2004, 96, 954–964, doi:10.1111/j.1365-2672.2004.02229.x.
c) Stoodley, P.; Sauer, K.; Davies, D.G.; Costerton, J.W. Biofilms as Complex Differentiated Communities. Annual review of microbiology 2002, 56, 187–209, doi:10.1146/annurev.micro.56.012302.160705.
d) Vázquez-Sánchez, D.; Rodríguez-López, P. Chapter 5 - Biofilm Formation of Staphylococcus Aureus. In; Fetsch, A.B.T.-S. aureus, Ed.; Academic Press, 2018; pp. 87–103 ISBN 978-0-12-809671-0
e) Staley, Z.R.; He, D.D.; Edge, T.A. Persistence of Fecal Contamination and Pathogenic Escherichia Coli O157:H7 in Snow and Snowmelt. Journal of Great Lakes Research 2017, 43, 248–254, doi:https://doi.org/10.1016/j.jglr.2017.01.006.
f) Maes, S.; Odlare, M.; Jonsson, A. Fecal Indicator Organisms in Northern Oligotrophic Rivers: An Explorative Study on Escherichia Coli Prevalence in a Mountain Region with Intense Tourism and Reindeer Herding. Environmental Monitoring and Assessment 2022, 194, 264, doi:10.1007/s10661-022-09865-1.
Reviewer 2 Report
Comments and Suggestions for Authors
The aim of this paper is to analyze culturable bacteriological indicators of water quality such as Escherichia coli, fecal enterococci, Salmonella, Staphylococcus), presence and concentration of antimicrobial agents, genes determining bacterial antibiotic resistance (ARGs), next-generation sequencing (NGS)-based bacterial community composition, and diversity in resource water and technical snow of five ski resorts. This is a very interesting topic. However, it lacks justification for why this huge amount of research was done. Reading the article, a reader might get the impression that the authors conducted research for the sake of conducting it, without a deeper analysis of why they are doing it. What is the purpose of this research? The authors do not draw any conclusions, do not point out any ways and opportunities to improve the situation they study. The authors also don’t indicate what sense it makes to perform this analysis for human health and the environment. The authors do not provide any guidelines for the future after analyzing such a huge amount of research. Nevertheless, there are some comments regarding the submitted manuscript:
Major comments
Line 25-31: The abstract should include summary information about the most important findings from the research. The abstract included information on what was studied, however, it lacked any conclusions that resulted from the research. It was written that changes occur without even introducing a plausible explanation for these changes. Instead, the authors wrote that this requires further research. Doing such a large amount of analysis and having as much data as the authors obtained, it is possible to come up with some conclusions. The absence of these conclusions makes the publication baseless. The authors need to look for correlations between the data obtained. There is also a lack of decent statistical analysis. Therefore, the authors are unable to make any conclusions from the results of this research. The paper needs to be supplemented with statistical calculations and demonstration of correlations between the data. The work must be completed with these data.
Although, the authors indicate important findings later in the article (after line 252), they did not draw important conclusions that should be included in the abstract. The authors must re-arrange the abstract to include appropriate conclusions from their research results.
In the introduction: The authors focused on describing new research techniques as the main goal of the paper, explaining that no such research has yet been done with these techniques. However, it should be emphasized that techniques, e.g. NGS, in the research world have already been used for several years therefore they are no novelty. They are well-described and well-known metagenomic techniques. So the use of NGS cannot be an end in itself. The authors need to write what relevance their research has had for humans and the environment, not focusing on the techniques but focusing on the significance of the research. The authors must describe what the problem is with artificial snowmaking, what health and environmental problems may result. These problems are hinted at in a few sentences at the beginning of the introduction, but the main part describes the techniques, which add nothing for the readers interested in the problem of artificial snowmaking. Besides, artificial snowmaking can also be done in countries such as Saudi Arabia, where there has been a lack of snow since time immemorial. So this is a general topic, not just related to one country and only climate changes. This should also be taken into consideration.
- Results and Discussion: The authors must compare their results to the literature data from other places in the word. For example, why is the amount of antibiotics high in some places and completely absent in others? What is the reason for this? What are the authors' assumptions in these specific places? etc. The authors meticulously described the results, but discuss them almost not at all. Authors must thoroughly discuss their research results. There are many studies of river water and some ones of snow hence there is a lot of data to discuss.
Conclusions: The authors should include the substantive conclusions of their research, as well as general conclusions based on result from the discussion with data from other places in the world. The authors should remove content that should be in the discussion section. In the conclusions section, the authors should also add some guidelines for ski resorts to make their artificial snow as safe as possible. This should include what these resorts need to do in the future. In this paper, the conclusions section contains a continuation of the discussion. The conclusion section should only contain conclusions and guidelines for the future. Please change this section.
Minor comments
Line 19 and line 65: If any name of a microorganism is used for the first time in the text, write the full generic name. Please write full names
Line 25: please expand on the acronym LOQ,
Line 29: potential pathogens to humans (Glutamicibacter). It have been very rarely described as causing disease in humans. The Glutamicibacter group of microbes is rather known for antibiotic and enzyme production. Under certain circumstances, practically any microorganism can be pathogenic, even probiotic Lactobacillus. So demonstrating a rare disease-causing microorganism is a weak finding. These bacteria are commonly found in soil, water, cheese, plants or clinical samples from humans. For this reason, the authors also discovered these bacteria. It should be noted that it is very rarely described as a pathogen, although it is very common in the environment. It is certainly not a clinically important pathogen as written in this paper. Please change it.
Line 92: Please expand on the acronyms: BDZ and BDF or explain that the region was given a particular code. At the moment it is not clear.
Ideally, the authors should split Table 1 into two tables. In the first table they would have included the codes of the regions with their description and latitude and longitude. The Supplement materials should include a map indicating the region codes. And in the second table they would include the codes of the regions with the results of microbiological purity of the water. This would have resulted in two smaller and more readable tables.
Table 2: The table should below the paragraph in which this table is mentioned. Please expand on the acronym LOQ under the table as explanation of the table
Line 106: Is this really Table 3? It seems that the authors were describing the results from Table 2.
Line 119: What is WWTPs?
Figure 2 should be placed after the text in which it is mentioned. Please move this Figure 2 after line 263
Line 468: please remove double dot.
Author Response
- This is a very interesting topic. However, it lacks justification for why this huge amount of research was done. Reading the article, a reader might get the impression that the authors conducted research for the sake of conducting it, without a deeper analysis of why they are doing it. What is the purpose of this research? The authors do not draw any conclusions, do not point out any ways and opportunities to improve the situation they study. The authors also don’t indicate what sense it makes to perform this analysis for human health and the environment. The authors do not provide any guidelines for the future after analyzing such a huge amount of research. Nevertheless, there are some comments regarding the submitted manuscript:
Response: First, we would like to sincerely thank the Reviewer for such thorough review of our manuscript, as well as detailed and justified remarks. We tried to answer all comments and improve the manuscript according to all suggestions.
And so, in response to this first general remark, after reading the mentioned sections, we agree that they lacked proper background of research and justification of undertaking this study. Thus, we thoroughly reorganized and rewritten some fragments of the text to meet the Reviewers’ suggestions. The Abstract has been almost completely rewritten, as well as the Introduction, from which we removed some unnecessary information (i.e. about the novelty of some techniques) and added solid reasons for undertaking the study. We reorganized and rewritten the aims of the study, provided more statistical analyses, and rewritten the conclusions, by removing the fragments that repeat the Discussion and organized our findings into a bullet list. We included some possible actions that might be taken by the ski station managers to improve the technical snow quality and safety of the environment and human health.
- Line 25-31: The abstract should include summary information about the most important findings from the research. The abstract included information on what was studied, however, it lacked any conclusions that resulted from the research. It was written that changes occur without even introducing a plausible explanation for these changes. Instead, the authors wrote that this requires further research. Doing such a large amount of analysis and having as much data as the authors obtained, it is possible to come up with some conclusions. The absence of these conclusions makes the publication baseless. The authors need to look for correlations between the data obtained. There is also a lack of decent statistical analysis. Therefore, the authors are unable to make any conclusions from the results of this research. The paper needs to be supplemented with statistical calculations and demonstration of correlations between the data. The work must be completed with these data.
Response: The Abstract was completely rewritten. The unnecessary fragments on e.g. continuation of research and methods description were shortened, while others (i.e. clearly stated aims, conclusions, recommendations) were added.
The statistical analysis of data in this study included: examination of data normality, followed by ANOVA and post-hoc tests to examine the significance of differences between the analyzed parameters, Pearson’s correlation analysis between all parameters and separately between ARGs and antibiotics, finally a multivariate statistical analysis, i.e. principal component analysis (PCA). These analyses were listed and described in the Materials and Methods section. Their detailed results are provided in the Supplementary Tables (S2 and S3), as well as they are mentioned and discussed throughout the manuscript text.
- Although, the authors indicate important findings later in the article (after line 252), they did not draw important conclusions that should be included in the abstract. The authors must re-arrange the abstract to include appropriate conclusions from their research results.
Response: the Abstract was rewritten to include conclusions and recommendations following from the study.
- In the introduction: The authors focused on describing new research techniques as the main goal of the paper, explaining that no such research has yet been done with these techniques. However, it should be emphasized that techniques, e.g. NGS, in the research world have already been used for several years therefore they are no novelty. They are well-described and well-known metagenomic techniques. So the use of NGS cannot be an end in itself. The authors need to write what relevance their research has had for humans and the environment, not focusing on the techniques but focusing on the significance of the research. The authors must describe what the problem is with artificial snowmaking, what health and environmental problems may result. These problems are hinted at in a few sentences at the beginning of the introduction, but the main part describes the techniques, which add nothing for the readers interested in the problem of artificial snowmaking. Besides, artificial snowmaking can also be done in countries such as Saudi Arabia, where there has been a lack of snow since time immemorial. So this is a general topic, not just related to one country and only climate changes. This should also be taken into consideration.
Response: The suggested fragments (novelty of the techniques used in the study) were removed. We added a few sections to the Introduction describing the background of undertaking this research, such as contamination of resource water by wastewater that results from rapid development of tourism, lack of removal of antibiotics by wastewater treatment plants (WWTPs) and the resulting spread of these agents, the impact antimicrobials have on antibiotic resistance and its determinants; we described the potential impact that spreading and dissemination of these micropollutants can have on human and environmental health.
We also added the fragment explaining why technical snowmaking and the associated issues are now becoming worldwide problems, due to the expected construction of large skiing complex in Trojena, Saudi Arabia and due to the development of all-weather snowmaking techniques that will allow for snowmaking independent of low temperatures. We also rewritten the aims and specified them.
- Results and Discussion: The authors must compare their results to the literature data from other places in the word. For example, why is the amount of antibiotics high in some places and completely absent in others? What is the reason for this? What are the authors' assumptions in these specific places? etc. The authors meticulously described the results, but discuss them almost not at all.
Authors must thoroughly discuss their research results. There are many studies of river water and some ones of snow hence there is a lot of data to discuss.
Response: We are not sure of the Reviewer meant the section that described the antibiotic content, as this section in fact thoroughly discusses the results, compares them with these obtained in other countries and suggests the possible explanations of the observed situation (i.e. in the corrected version: lines 162-169, 178-180; 180-194 if the changes are hidden and lines 193-200, 201-211, 221-234 if the changes are highlighted).
However, if the Reviewer meant the section referring to the Culture-based assessment of bacteriological contamination, we indeed haven’t discussed the results which was an obvious mistake. This has been corrected and the results have been discussed by comparing to these from other parts of the world as well as by an attempt at explaining the observed differences (l. 119-144 of the manuscript with hidden changes and l. 139-170 with highlighted changes).
We also removed or shortened fragments where the results were described in details and referred to the respective tables and/or figures.
- Conclusions: The authors should include the substantive conclusions of their research, as well as general conclusions based on result from the discussion with data from other places in the world. The authors should remove content that should be in the discussion section. In the conclusions section, the authors should also add some guidelines for ski resorts to make their artificial snow as safe as possible. This should include what these resorts need to do in the future. In this paper, the conclusions section contains a continuation of the discussion. The conclusion section should only contain conclusions and guidelines for the future. Please change this section.
Response: The Conclusions were also thoroughly reorganized by:
- a) removing the fragments that repeat the discussion;
- b) systematizing the observations and organizing the findings into bullet point list
- c) providing guidelines that the ski stations may implement to improve the safety of technical snow they produce in case of resource water contamination.
Minor
- Line 19 and line 65: If any name of a microorganism is used for the first time in the text, write the full generic name. Please write full names
Response: the bacterial names first mentioned in the text have been provided in full names
- Line 25: please expand on the acronym LOQ,
Response: the acronym has been explained
- Line 29: potential pathogens to humans (Glutamicibacter). It have been very rarely described as causing disease in humans. The Glutamicibacter group of microbes is rather known for antibiotic and enzyme production. Under certain circumstances, practically any microorganism can be pathogenic, even probiotic Lactobacillus. So demonstrating a rare disease-causing microorganism is a weak finding. These bacteria are commonly found in soil, water, cheese, plants or clinical samples from humans. For this reason, the authors also discovered these bacteria. It should be noted that it is very rarely described as a pathogen, although it is very common in the environment. It is certainly not a clinically important pathogen as written in this paper. Please change it.
Response: Information on Glutamicibacter has been removed from the abstract and changed in the main text as suggested.
- Line 92: Please expand on the acronyms: BDZ and BDF or explain that the region was given a particular code. At the moment it is not clear.
Response: information was added that the explanation of acronyms is provided in Materials and Methods in the table
- Ideally, the authors should split Table 1 into two tables. In the first table they would have included the codes of the regions with their description and latitude and longitude. The Supplement materials should include a map indicating the region codes. And in the second table they would include the codes of the regions with the results of microbiological purity of the water. This would have resulted in two smaller and more readable tables.
Response: The Table 1 was split into two tables as suggested. One (Table 1) was placed in Results and Discussion and contains the numerical results and abbreviations, the other one (Table 4) was moved to Materials and Methods and explains the location, height, population, anthropogenic pressure and explains abbreviations.
The map presenting the location of examined regions was added into Supplementary materials.
- Table 2: The table should below the paragraph in which this table is mentioned. Please expand on the acronym LOQ under the table as explanation of the table
Response: Table 2 was already placed below the paragraph that described its results (i.e. mean concentrations of antimicrobial agents). LOQ was explained in the table caption
- Line 106: Is this really Table 3? It seems that the authors were describing the results from Table 2.
Response: corrected. Indeed, this should have been Table 2.
- Line 119: What is WWTPs?
Response: The abbreviation has been explained
- Figure 2 should be placed after the text in which it is mentioned. Please move this Figure 2 after line 263
Response: done as suggested.
- Line 468: please remove double dot.
Response: done
Round 2
Reviewer 2 Report
Comments and Suggestions for Authors
The authors have improved their paper very well. Now, the article reads with great interest. It can be very helpful for all snowmaking services around the world.
I have only two minor comments:
Line 153: This is a publication for people from all over the world, even for those who do not know geography well therefore after „Tatra National Park” please add “, in Poland and Slovakia, Europe”
Table 1: The authors should insert Table 1 at the end of the paragraph where they mention this table for the first time. Thus, please insert Table 1 after line 157.
Author Response
Dear Reaviewer,
Thank you for readinig the corrected manuscript and further suggestions.
We incorporated the suggested changes:
- Line 153: This is a publication for people from all over the world, even for those who do not know geography well therefore after „Tatra National Park” please add “, in Poland and Slovakia, Europe”
Response: done.
2. Table 1: The authors should insert Table 1 at the end of the paragraph where they mention this table for the first time. Thus, please insert Table 1 after line 157.
Response: moved as suggested.
We hope that the Reviewer and Editor will find the corrected version of the manuscript acceptable for publication.
With kind regards,
Anna Lenart-Boroń, corresponding author